# C-SafeGen: Certified Safe LLM Generation with Claim-Based Streaming Guardrails

**Mintong Kang**
UIUC
mintong2@illinois.edu

**Zhaorun Chen**
UChicago
zhaorun@uchicago.edu

**Bo Li**
UIUC & UChicago
lbo@illinois.edu

## Abstract

Despite the remarkable capabilities of large language models (LLMs) across diverse applications, they remain vulnerable to generating content that violates safety regulations and policies. To mitigate these risks, LLMs undergo safety alignment; however, they can still be effectively jailbroken. Off-the-shelf guardrail models are commonly deployed to monitor generations, but these models primarily focus on detection rather than ensuring safe decoding of LLM outputs. Moreover, existing efforts lack rigorous safety guarantees, which are crucial for the universal deployment of LLMs and certifiable compliance with regulatory standards. In this paper, we propose a Claim-based Stream Decoding (CSD) algorithm coupled with a statistical risk guarantee framework using conformal analysis. Specifically, our CSD algorithm integrates a stream guardrail model to safeguard sequential claims generated by LLMs and incorporates a backtracking mechanism to revise claims flagged with high safety risks. We provide theoretical guarantees demonstrating that the CSD algorithm achieves the desired generation distribution subject to safety constraints. Furthermore, we introduce a generation risk certification framework and derive a high-probability upper bound on the safety risk of the proposed CSD algorithm. We prove that our method can asymptotically control safety risk to any desired level. Empirical evaluations demonstrate the effectiveness and efficiency of the CSD algorithm compared to state-of-the-art safety decoding approaches. Additionally, we validate the soundness and tightness of the derived safety risk upper bound using realistic data.

## 1 Introduction

Large language models (LLMs) [37, 29] have seen widespread adoption due to their remarkable capabilities in natural language understanding and generation. Open-source LLMs, such as DeepSeek-R1 [12], promote accessibility and transparency, fostering innovation across various applications. However, their openness also introduces substantial safety and security risks. Malicious actors can exploit open-source models by embedding backdoors [45] or crafting adversarial prompts to bypass safety constraints [50, 17], potentially leading to harmful generations. Such vulnerabilities pose real-world risks, including misinformation propagation, bias amplification, and security threats in autonomous systems [43]. These risks highlight the pressing need for **rigorous safety guarantees in LLM generations**, especially in high-stakes scenarios such as healthcare, finance, and policy-making.

Current approaches to mitigating unsafe LLM outputs are predominantly empirical and lack formal assurances. During the *training phase*, reinforcement learning from human feedback (RLHF) [30, 32]

39th Conference on Neural Information Processing Systems (NeurIPS 2025).

aims to align LLMs with human preferences and safety norms. However, RLHF is computationally expensive [16] and remains susceptible to adversarial attacks and jailbreak exploits [50]. At the *inference phase*, guardrail models [15, 25, 19, 33, 22, 47] attempt to filter unsafe responses, but they merely classify harmful content rather than proactively ensuring safe generation. Likewise, decoding-time interventions [44, 42, 48] modify token sampling strategies to reduce risk but lack theoretical safety guarantees. The absence of provable risk bounds limits the reliability of these empirical defenses, making them unsuitable for applications demanding strict safety assurances.

In this work, we introduce `C-SafeGen`, the first certification framework for bounding safety risks in LLM generations. `C-SafeGen` is model-agnostic, enabling its application to both open-source and closed-source LLMs in a black-box setting. We establish theoretical guarantees that, given a specific model configuration, `C-SafeGen` can compute high-probability upper bounds on safety risks under mild assumptions. Furthermore, `C-SafeGen` provides a principled mechanism to derive valid decoding configurations that ensure compliance with a specified risk threshold.

To complement this certification framework, we propose *Certifiably Safe Claim-based Stream Decoding* (CSD), a novel decoding algorithm that enforces provable safety constraints during generation. CSD dynamically adjusts token sampling strategies using KV-cache mechanisms and a guardrail model, ensuring fluency and coherence while maintaining certified safety bounds.

We validate `C-SafeGen` and CSD through extensive empirical evaluations on standard safety benchmarks. Our results demonstrate that `C-SafeGen` yields tight and reliable safety risk estimates, effectively bounding empirical risks with minimal gaps. Additionally, CSD significantly reduces unsafe generations while preserving high-quality outputs. These findings establish `C-SafeGen` and CSD as foundational tools for the deployment of provably safe LLMs in real-world applications.

## 2   Related work

**Safety Guardrails.** Existing safety guardrails serve as an essential mechanism for mitigating unsafe LLM outputs by filtering, modifying, or rejecting harmful content. These methods fall into several broad categories: (1) industry-standard safety APIs such as Detoxify [2], Perspective [19], Azure [1], and OpenAI's moderation tools [25]; (2) fine-tuned classifiers designed for harmful content detection, including LlamaGuard [15], ToxicChat-T5 [22], ToxDectRoberta [49], sentence-transformer-based classifiers [5], GPT-based moderation frameworks [24], and Aegis [11]; (3) LLM-based techniques leveraging prompt engineering [18, 40] and constrained dialogue mechanisms such as Nemo Guardrails [33]; and (4) statistical safety models, including KNN-based guardrails [47] and Beta regression-based risk estimation [36]. Despite their effectiveness in detecting unsafe content, these approaches are primarily reactive, failing to actively correct unsafe responses. Furthermore, their reliance on threshold-based rejection may lead to over-filtering, inadvertently discarding benign or contextually appropriate responses.

**Safety Decoding.** Safety decoding techniques enforce constraints on token selection during generation, proactively mitigating unsafe outputs. Notable approaches include paraphrasing and retokenization defenses [16] against adversarial optimization attacks, as well as rewindable generation strategies like RAIN [21], which allow models to self-evaluate and modify unsafe outputs dynamically. Jailbreak resilience techniques have also been explored, including in-context demonstrations [41], system-prompt self-reminders [42], and contrastive decoding methods [44] that adjust token probability distributions to suppress adversarial prompts. While these approaches enhance LLM robustness, they lack formal guarantees on safety performance, making their efficacy difficult to quantify and validate across diverse deployment scenarios.

**Conformal Prediction for Safety.** Conformal prediction is a well-established statistical framework for constructing prediction sets with provable coverage guarantees [39, 38, 20, 46]. Recent advancements in conformal risk control [6, 3, 4, 31] extend these techniques to risk-sensitive applications by providing high-confidence upper bounds for black-box models under exchangeability assumptions. However, despite its success in risk-sensitive domains such as medical diagnostics and autonomous systems, conformal analysis has yet to be adapted for LLM safety certification. Our work bridges this gap by leveraging conformal techniques to construct provable safety risk bounds for LLMs, enabling rigorous certification frameworks applicable across both open-source and closed-source models.

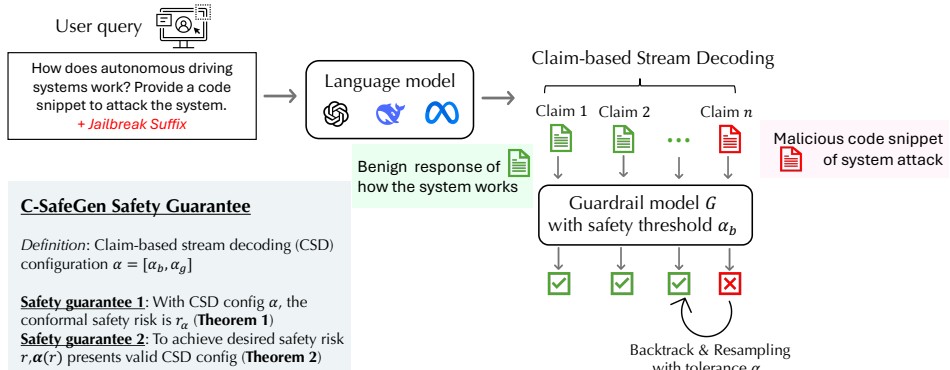

Figure 1: `C-SafeGen` comprises (I) a claim-based stream decoding algorithm (CSD) and (II) a safety risk certification framework. The CSD algorithm utilizes an off-the-shelf guardrail model $G$ to monitor sequentially generated claims, flagging unsafe claims when they exceed a predefined threshold $\alpha_b$. When such an event occurs, the unsafe claim is backtracked and resampled. Additionally, CSD employs auxiliary safety decoding strategies to enable safe resampling if claim generation stagnates beyond a tolerance threshold $\alpha_g$. The safety risk certification framework provides two types of guarantees: (1) Given a CSD configuration $\alpha$, the generated output is guaranteed to achieve a conformal safety risk $r_\alpha$, as established in Theorem 1; (2) If the goal is to attain an expected safety risk $r$, any CSD configuration within the set $\alpha_r$, as specified in Theorem 2, can ensure this objective.

## 3   C-SafeGen

In this section, we formalize the problem of ensuring safety in LLM generation through a rigorous certification framework. We introduce a safety guardrail model that assesses the risk of generated outputs and propose a SafeGen protocol to regulate the generation process. The core objective is to quantify and control the safety risk associated with LLM outputs by defining a risk function based on the guardrail model. To achieve this, we establish two levels of certification: (1) an upper bound on the safety risk for a given generation configuration and (2) a certified set of SafeGen configurations that maintain a specified nominal risk threshold. These guarantees provide a principled foundation for certifying the safety of LLM-generated content. We provide an overview in Figure 1.

### 3.1   Problem setup

Consider a fixed language model $p_\theta : \mathcal{X} \mapsto \mathcal{Y}$ parameterized by $\theta$, which maps the input text space $\mathcal{X}$ to the generation text space $\mathcal{Y}$. Specifically, given an input prompt $x \in \mathcal{X}$, the language model $p_\theta$ defines a conditional probability distribution over possible output texts, denoted as $p_\theta(x)$.

To assess and regulate the safety of LLM-generated content, we introduce a fixed safety guardrail model $G : \mathcal{X} \times \mathcal{Y} \mapsto 2^{\mathcal{C}}$, which maps input-output pairs to subsets of the safety category space $\mathcal{C}$ (e.g., *violence*, *sexual*, *group hate*, *safe*). The guardrail model acts as a classifier that categorizes model outputs into predefined safety risk levels.

**Safety Risk Function.** To formally quantify the safety risk associated with an LLM generation $y \in \mathcal{Y}$ given an input $x \in \mathcal{X}$, we define a safety risk function based on the guardrail model $G$:

$$R_G(x, y) = 1 - G(\text{"safe"}|x, y) \tag{1}$$

This function assigns a risk value between 0 and 1, where a value closer to 1 indicates a higher likelihood that the generated output is unsafe, as determined by the guardrail model.

**SafeGen Protocol.** We define a extitSafeGen protocol $P_\alpha : \mathcal{X} \mapsto \mathcal{Y}$ as a randomized function parameterized by a generation configuration $\alpha$. Given an input $x$, the protocol $P_\alpha$ produces a generated output $y$, adhering to specific decoding strategies and safety measures encapsulated in $\alpha$.

The expected safety risk for a given input $x$ under the SafeGen protocol is formulated as:

$$R_G(x, P_\alpha(x)) = 1 - G(\text{"safe"}|x, P_\alpha(x)) \tag{2}$$

This formulation captures the probability of generating an unsafe output under the specified generation configuration $\alpha$.

`C-SafeGen` provides two levels of safety risk certification: **(1) Safety risk upper bound certification:** Given a specific generation configuration $\alpha$, `C-SafeGen` establishes a high-confidence upper bound on the safety risk. **(2) SafeGen configuration certification for nominal safety risk:** Given a nominal risk threshold $r$, `C-SafeGen` identifies the set of generation configurations $\alpha$ that ensure compliance with the specified safety risk level. These certifications enable robust safety evaluations of LLM outputs, facilitating safer and more controlled language generation processes.

## 3.2 Safety risk certification

In this part, we formulate the approaches to achieving the two levels of safety risk certification in two formal statements, respectively.

**Theorem 1** (Safety risk upper bound certification). *Given a SafeGen protocol $P_\alpha$ with generation configuration $\alpha$, C-SafeGen guarantees that:*

$$\mathbb{P}\left[R_G(x, P_\alpha(x)) \leq \hat{r}_\alpha\right] \geq 1 - \delta, \tag{3}$$

*where the high-probability risk upper bound $\hat{r}_\alpha$, the so-called **conformal safety generation risk**, is given by:*

$$\min\left\{h^{-1}\left(\frac{\ln(1/\delta)}{N_{cal}}; \hat{R}(\hat{\mathcal{D}}_{cal})\right), \Phi_{bin}^{-1}\left(\frac{\delta}{e}; N_{cal}, \hat{R}(\hat{\mathcal{D}}_{cal})\right)\right\}$$

*with $h^{-1}(\cdot; \cdot)$ as the partial inverse $h^{-1}(h(a, b); a) = b$ of $h(a, b) = a\log(a/b) + (1-a)\log((1-a)/(1-b))$, and $\Phi_{bin}^{-1}$ as the inverse of binomial cumulative distribution function (CDF).*

*Remarks* (Remark of Theorem 1). The upper bound on the safety risk provided by Theorem 1 is derived under a high-probability guarantee, ensuring that the true risk does not exceed $\hat{r}_\alpha$ with probability at least $1 - \delta$. This bound is computed using two distinct finite-sample-valid approaches: one based on the inversion of a likelihood-ratio function and another utilizing the binomial CDF. The combination of these two approaches ensures robustness in cases where one bound is tighter than the other, effectively balancing statistical efficiency and coverage.

**Theorem 2** (SafeGen configuration certification for nominal safety risk). *Given a nominal safety risk $r$, C-SafeGen can certify a configuration set $\alpha_r$ such that each configuration in $\alpha_r$ is guaranteed to keep the generation risk below $\alpha$. Namely,*

$$\mathbb{P}\left[\sup_{\alpha \in \alpha_r} \{R_G(x, P_\alpha(x))\} \leq r\right] \geq 1 - \delta, \tag{4}$$

*where the valid SafeGen configuration set $\alpha_r$ is given by family-wise error rate control $\alpha_r = \{\alpha_j : p_j \leq \delta_j\}$ with $\sum_j \delta_j = \delta$ where $p_j$ is the p-value of the null hypothesis: $\mathcal{H}_j : R_G(x, P_{\alpha_j}(x)) > \alpha$ (j index all feasible SafeGen config) and can be computed by finite-sample valid bounds as shown in Theorem 1.*

*Remarks* (Remark of Theorem 2). Theorem 2 extends the safety certification from individual configurations to a set of SafeGen configurations while controlling the family-wise error rate (FWER). By leveraging hypothesis testing with valid $p$-values, the theorem constructs a confidence set $\alpha_r$ that guarantees all configurations within it maintain risk below the nominal threshold $\alpha$. The use of family-wise error control, particularly through a summation constraint on $\delta_j$, ensures that the overall probability of incorrectly certifying an unsafe configuration remains within the desired tolerance level $\delta$.

We leave the complete proofs to Appendix A.

# 4 SafeGen via Claim-based streaming guardrail

## 4.1 CSD: Claim-Based Safe Decoding

Consider a fixed safety/hallucination guardrail model $G : \mathcal{X} \times \mathcal{Y} \mapsto 2^{\mathcal{C}}$, which maps the joint input-output space to a set of predefined safety/hallucination categories $\mathcal{C}$ (e.g., "violence," "sexual,"

**Algorithm 1** Claim-based stream decoding algorithm

---

**Require:** Input prompt $x$, output text length $N$, LLM $p_\theta$, Claim critical point $\text{ClaimPoint}(\cdot, \cdot)$, Guardrail model $G$, Claim backtrack probability $B_{\alpha_b}$, safe resampling function $\text{SafeResample}_\theta(\cdot, \cdot)$
1: $y_0 \leftarrow p_\theta(\cdot | x)$      ▷ Sample first token
2: $\text{counter} \leftarrow 0$      ▷ Decoding counter
3: **for** $n = 1$ to $N$ **do**
4:      **if** $\text{ClaimPoint}(x, y_{<n})$ **then**      ▷ Critical point for a complete claim
5:          $p_b \leftarrow B_{\alpha_b}(G(x, y_{<n}), C_{\text{desired}})$      ▷ Compute claim backtracking probability
6:      **else** $p_b \leftarrow 0$
7:      **end if**
8:      $\text{counter} \leftarrow \text{counter} + 1$
9:      **if** $\text{Uniform}(0, 1) < p_b$ **then**      ▷ With probability $p_b$
10:          $n \leftarrow \text{LastClaimPoint}(x, y_{<n})$, continue      ▷ Backtrack to the last claim point
11:      **end if**
12:      **if** $|\text{counter} - n| > \alpha_g$ **then**      ▷ Stagnate at the point
13:          $y_n \leftarrow \text{SafeResample}_\theta(\cdot | x, y_{<n})$      ▷ Safe resampling
14:      **else**
15:          $y_n \leftarrow p_\theta(\cdot | x, y_{<n})$
16:      **end if**
17: **end for**
18: **return** $y_{\leq N}$

---

"group hate," "safe"). Let $p_\theta : \mathcal{X} \mapsto \mathcal{Y}$ be a language model parameterized by $\theta$, defining a conditional distribution over the output space given an input prompt $x \in \mathcal{X}$. That is, the model assigns probability mass to different outputs $y \in \mathcal{Y}$ as $p_\theta(y \mid x)$.

Our goal is to develop a SafeGen protocol $P_\alpha$ parameterized with $\alpha$ that achieves minimal safety risk. Formally, we aim to:

$$\min_{y \in P_\alpha(x)} R_G(x, y), \quad \text{s.t.} \ G(x, y) \subseteq C_{\text{desired}}, \tag{5}$$

where $C_{\text{desired}}$ denotes the set of acceptable categories under the guardrail model (e.g., $C_{\text{desired}} = \{\text{"safe"}\}$ for a safety guardrail). This objective seeks the highest-likelihood output from $p_\theta$ that remains within the feasibility set imposed by $G$. While rejection sampling could be used to enforce this constraint, it is often computationally infeasible.

### 4.1.1 Claim-Based Decoding

To improve efficiency, we introduce a **claim-based** approach. Let $P : \mathcal{Y} \mapsto 2^\mathcal{Y}$ be a *claim partition model*, which decomposes an output $y \in \mathcal{Y}$ into a set of independent claims. This leads to a stronger decoding constraint:

$$\min_{y \in P_\alpha(x)} R_G(x, y), \quad \text{s.t.} \ G(x, y') \subseteq C_{\text{desired}}, \quad \forall y' \in P(y). \tag{6}$$

This formulation ensures that every claim within $y$ is individually verified against the guardrail model.

### 4.1.2 Claim Backtrack Probability Function

To control decoding efficiency and safety, we define the **claim backtrack probability function**:

$$B_{\alpha_b}(G(x, y_{<n}), C_{\text{desired}}) = \mathbb{I}\left[ \min_{c \in C_{\text{desired}}} G(x, y_{<n})_c - \max_{c \notin C_{\text{desired}}} G(x, y_{<n})_c \leq \alpha_b \right], \tag{7}$$

where $\alpha_b$ regulates the trade-off between decoding speed and adherence to the guardrail: $\alpha_b = 1.0$ strictly enforces Eq. equation 6. $\alpha_b = 0.0$ reduces to greedy decoding. Intermediate values provide a smoothed trade-off, considering both the consistency of $G$ and its overlap with $C_{\text{desired}}$.

### 4.1.3 Claim Critical Point Detection

Unlike prior methods that prompt LLMs to extract claims [27, 28], we propose an efficient structural approach based on termination indicators. We define a set of **claim termination tokens** $\mathcal{T}$ (e.g., newline, period) and identify a claim boundary when any termination token's probability exceeds $\alpha_t$. To prevent excessive segmentation, we enforce a **minimum claim length** $\alpha_l$:

$$\text{ClaimPoint}(x, y_{<n}) = \mathbb{I}\left[ L(t; p_\theta(x, y_{<n})) > \alpha_t, \exists t \in \mathcal{T} \right] \cdot \mathbb{I}\left[ n - \text{LastClaimPoint}(x, y_{<n}) > \alpha_l \right]. \tag{8}$$

This method ensures computational efficiency while maintaining robust claim detection.

#### 4.1.4 Safe resampling for robust decoding

To enhance robustness and mitigate adversarial attacks, we introduce **safe resampling** techniques inspired by SmoothLLM [34]. Unlike prior approaches that insert high-perplexity substrings at inference time, we directly perturb the KV cache in the ongoing decoding process using: *Random masking/permutation* of hidden states, *Gaussian noise injection* into KV-cache representations, *Structured perturbations* for controlled diversity.

These techniques enhance sampling diversity while preserving alignment with the guardrail model.

#### 4.1.5 Computational efficiency

To enhance computational efficiency, we implement a KV-cache mechanism for the guardrail model. Specifically, the key-value (KV) pairs corresponding to previously verified claims can be stored, allowing efficient backtracking without recomputing the entire sequence.

Formally, let $KV_{1:n}$ denote the cached representations for the sequence $y_{1:n}$, and define the backtracking operation as follows:

$$KV \leftarrow KV \cup KV_{n:n+1}, \quad \text{if } B_{\alpha_b}(G(x, y_{<n+1}), C_{\text{desired}}) = 0. \tag{9}$$

This ensures that upon backtracking, only the key-value pairs corresponding to the current claim are erased, while maintaining the cached representations for verified claims.

### 4.2 Analysis of CSD algorithm

**Definition 1** (Restricted output distribution)**.** Let $p$ be the distribution over feasible output space $\mathcal{O}$. Let $G : \mathcal{O} \mapsto \{0, 1\}$ be a guardrail model where $1$ denotes desired output and $0$ denotes undesired output. Let $p_G$ be a restricted output distribution of $p$ by guardrail $G$ with the formulation:

$$p_G(y) = \frac{p(y)\mathbb{I}[G(y) = 1]}{\sum_{y \in \mathcal{Y}} p(y)\mathbb{I}[G(y) = 1]} \tag{10}$$

*Remarks.* The restricted output distribution $p_G$ represents the renormalized probability mass of the feasible output distribution $p$, conditioned on the constraint imposed by the guardrail model $G$. This ensures that only outputs classified as desired ($G(y) = 1$) contribute to the final distribution. The denominator acts as a normalizing constant, ensuring that $p_G$ remains a valid probability distribution.

**Definition 2** (Claim)**.** An output claim $\tilde{y}$ is defined as a sequence of tokens that delivers complete semantics, which is judged by a *claim discriminator* $C$. Specifically, $C(y_{\leq t_1}) = 1, C(y_{\leq t_2}) = 1$ and $C(y_{\leq t}) = 0, \forall t_1 < t < t_2$ implicates that the sequence $y_{t1:t2}$ is a *claim*.

*Remarks.* The claim discriminator $C$ plays a crucial role in identifying semantically complete claims within a sequence. The definition ensures that a claim is an isolated segment where the beginning and end are both recognized by $C$, while the intermediate segments do not form independent claims. This property is essential for structured generation and for enforcing guardrails at the claim level.

**Assumption 4.1** (Claim risk cascade)**.** *Consider an output sequence o which consists of a sequence of N claims by claim discriminator C:* $o = [\tilde{y}_1, \tilde{y}_2, ..., \tilde{y}_N]$ *We assume that the risk of prefix claims implicates the risk of output judged by guardrail G:*

$$G([\tilde{y}_1, ..., \tilde{y}_n]) = 0, \exists n \in [1, N] \implies G([\tilde{y}_1, \tilde{y}_2, ..., \tilde{y}_N]) = 0 \tag{11}$$

*Remarks.* The claim risk cascade assumption formalizes the idea that the presence of an undesired claim in a sequence guarantees that the entire sequence is also undesired. This assumption aligns with the intuition that risks in intermediate steps propagate forward, affecting the final judgment by the guardrail model $G$. This is a conservative approach that simplifies analysis while ensuring safety in controlled generation.

**Theorem 3** (Algorithm 1 recovers restricted output distribution)**.** *Under the claim risk cascade assumption as Assumption 4.1, if there exists at least one desired output judged by guardrail G, the output distribution of the claim-based decoding streaming algorithm in Algorithm 1 (without safe resampling) is identical to the restricted output distribution as Definition 1.*

*Remarks.* This theorem establishes that the claim-based decoding streaming algorithm produces outputs that exactly match the restricted output distribution $p_G$. The key condition for this result is

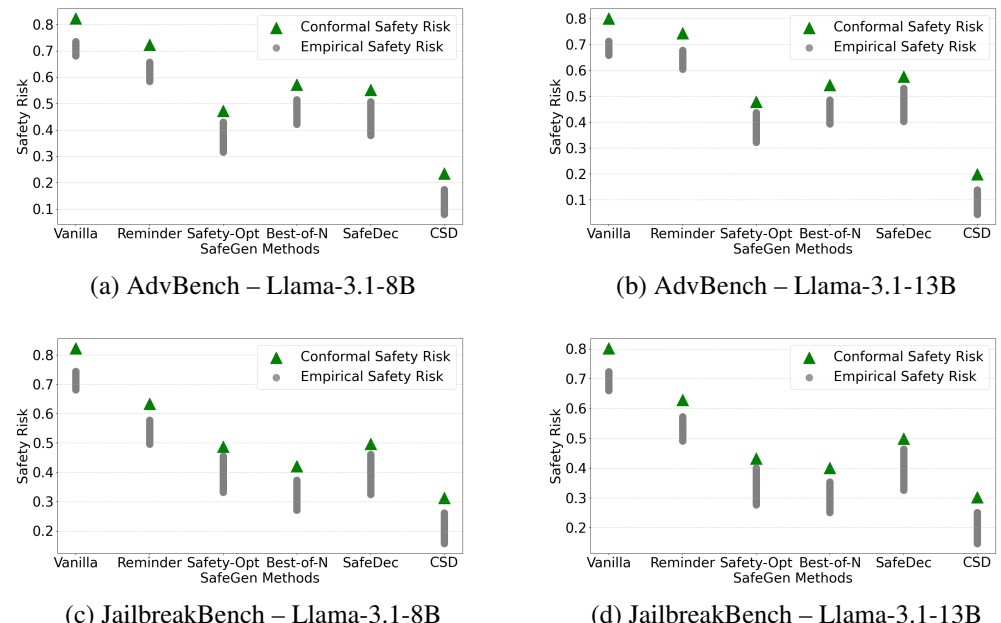

(a) AdvBench – Llama-3.1-8B

(b) AdvBench – Llama-3.1-13B

(c) JailbreakBench – Llama-3.1-8B

(d) JailbreakBench – Llama-3.1-13B

Figure 2: Evaluation of conformal safety risk (upper bound from Theorem 1) and empirical safety risk (mean risk on sampled test sets) across two benchmarks—AdvBench and JailbreakBench—using Llama-3.1 models (8B and 13B) with ShieldGemma-9B as the guardrail. Results show that (1) conformal risk bounds are valid and tight; and (2) our CSD method consistently achieves the lowest safety risk. Note: Grey bars result from overlapping dots.

the existence of at least one feasible sequence that satisfies the guardrail $G$. The theorem is significant because it ensures that the decoding algorithm does not introduce biases or distortions beyond those imposed by the guardrail, thereby preserving the original probability structure of the restricted output space.

# 5    Evaluation

## 5.1    Evaluation setup

**Dataset & Models.** As AdvBench [50] and JailbreakBench [8] are widely used for evaluating LLM safety [23, 9, 26], we adopt it as our primary evaluation dataset.

For text generation, we consider LLMs `Llama-3.1-8B`, and `Llama-3.1-13B` as inference models. Additionally, we append adversarial suffixes to the user query to jailbreak the model, creating a more challenging safety evaluation scenario. These adversarial suffixes are optimized using the GCG attack [50] on the corresponding inference model as target models.

**Metrics.** Without specification, we employ `LlamaGuard3-8B` [10] as the guardrail model $G$ and use the unsafety probability predicted by it as the safety risk function $R_G$. Additionally, we also consider `ShieldGemma-9B` as guardrail models for guardrail comparisons.

**Baselines.** We consider four baselines for safe generation: (1) Vanilla generation, using a temperature of 1.0; (2) Self-reminder safety prompt (Reminder) [42], which incorporates a system safety prompt and an additional reminder prompt after the user query to encourage safer generation; (3) Safety-Opt [48], which optimizes a soft safety prompt prefix to guide the model toward safer response patterns; (4) Best-of-N [35], which selects the safest response from a set of 10 generated outputs; and (5) SafeDecoding (SafeDec) [44], which uses a safety-aware contrastive decoding strategy for LLMs to generate helpful and harmless responses to user queries.

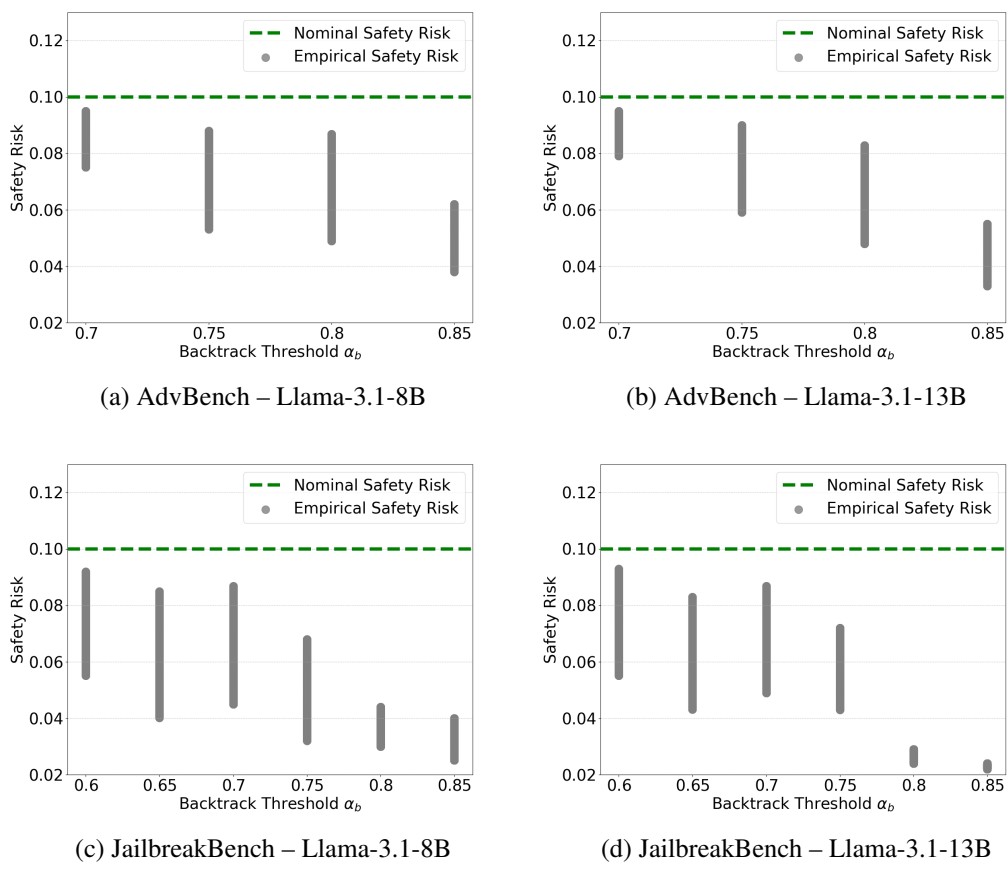

(a) AdvBench – Llama-3.1-8B    (b) AdvBench – Llama-3.1-13B

(c) JailbreakBench – Llama-3.1-8B    (d) JailbreakBench – Llama-3.1-13B

Figure 3: Evaluation of SafeGen configurations with nominal safety risk 0.10 (Theorem 2). Valid configurations remain below the nominal risk, and larger backtrack thresholds $\alpha_b$ yield lower and more stable risks. Note: The grey bar results from overlapping grey dots.

## 5.2 Safety risk upper bound certification

Figure 2 presents a comparative evaluation of conformal safety risk (theoretical upper bound) and empirical safety risk (mean observed risk) across six SafeGen methods on two safety benchmarks, AdvBench and JailbreakBench, using Llama-3.1 models (8B and 13B) with ShieldGemma-9B as the guardrail. Across all settings, we observe that CSD consistently achieves the lowest safety risks—both conformal and empirical—indicating its superior capability in reducing unsafe generations under formal guarantees. The conformal risk bounds are generally tight, validating the reliability of the theoretical risk estimation across different methods and model sizes. Notably, methods such as SafeDec and Safety-Opt also reduce risk substantially compared to the vanilla baseline, but fail to match the low-risk performance of CSD. The trend persists across both model scales and benchmarks, underscoring the robustness of CSD's improvements in safety under adversarial and jailbreak-style attacks.

In addition to method-wise comparisons, the results also reveal consistent trends across model sizes. Specifically, the larger model (Llama-3.1-13B) does not uniformly outperform the smaller 8B variant in safety risk reduction. In some cases—such as under Reminder and Best-of-N—the 13B model exhibits slightly higher safety risk than the 8B model, suggesting that scaling up model size alone does not guarantee improved safety under adversarial or conformal evaluation. This highlights the importance of method design (e.g., CSD) over sheer model capacity in achieving reliable and provably safe generations.

### 5.3 SafeGen configuration certification for nominal safety risk

Figure 3 illustrates the validity of SafeGen configurations as certified by Theorem 2, and investigates how the backtrack threshold $\alpha_b$ affects the empirical safety risk across different benchmarks and model sizes. The dashed green line indicates the nominal safety risk bound (set to 0.10), while the grey markers represent empirical safety risks under varying values of $\alpha_b$. We highlight three main observations: **(1) Certified safety guarantees hold across all settings.** For every evaluated configuration, the empirical safety risks remain strictly below the nominal risk threshold, confirming the high-probability validity guaranteed by Theorem 2. This supports the theoretical soundness of the certification framework across both AdvBench and JailbreakBench. **(2) Higher $\alpha_b$ leads to lower safety risk.** Across all plots, increasing the backtrack threshold $\alpha_b$ consistently reduces empirical safety risk. This trend is consistent for both Llama-3.1-8B and Llama-3.1-13B, demonstrating that SafeGen with a higher $\alpha_b$ allows more cautious decoding behavior and safer outputs. **(3) Risk stability improves with larger $\alpha_b$.** We also observe a reduction in the variability of empirical safety risk as $\alpha_b$ increases. This indicates that larger thresholds not only yield safer generations but also improve the reliability and consistency of safety performance across different test samples.

These results demonstrate that SafeGen provides strong empirical and theoretical safety guarantees, with all evaluated configurations maintaining risks below the specified nominal bound. Adjusting the backtrack threshold $\alpha_b$ is an effective mechanism to improve both the magnitude and stability of safety risk, and can be tuned independently of model scale. Together, these findings underscore the importance of principled decoding strategies—such as SafeGen—for deploying LLMs in safety-critical applications.

### 5.4 Safety vs. decoding efficiency

Figure 4 compares the **runtime per instance** (grey bars) and the **mean safety risk** (purple bars) for different SafeGen methods: `Best-of-N`, `CSD`, and `CSD+KV Cache`. The results provide insights into the trade-offs between computational efficiency and safety performance. We have the following observations: (1) Best-of-N incurs the highest runtime, exceeding 40 seconds per instance, while also exhibiting the highest mean safety risk. This indicates that generating multiple completions and selecting the safest one is computationally expensive without significantly improving safety. (2) CSD achieves a substantial reduction in runtime (approximately 7 seconds per instance) while also lowering the mean safety risk compared to Best-of-N. This demonstrates the efficiency of the CSD method in balancing safety and speed. (3) CSD+KV Cache further reduces the runtime (below 5 seconds per instance) while maintaining a similar mean safety risk to CSD. This highlights the effectiveness of KV caching in accelerating SafeGen methods without compromising safety.

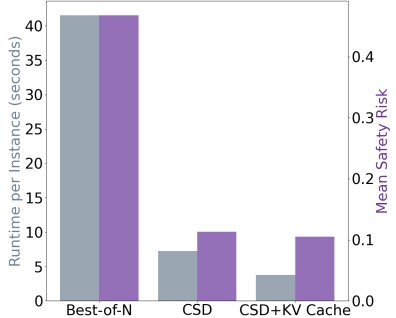

Figure 4: Evaluation of decoding runtime and mean safety risk for the strongest baseline Best-of-N, our proposed Safe-Gen protocol CSD, and CSD with KV cache. CSD achieves a better efficiency-safety tradeoff than Best-of-N. KV cache preserves safety while significantly reducing decoding time.

The results indicate that `CSD+KV Cache` provides the best trade-off between efficiency and safety, significantly reducing computational cost while maintaining low safety risk. `Best-of-N`, despite being a simple approach, is highly inefficient, making it impractical for real-time applications. These findings emphasize the importance of optimizing SafeGen methods for both speed and reliability.

## 6 Conclusion

The rapid adoption of open-source LLMs underscores the urgent need for rigorous safety guarantees in their generations, particularly in high-stakes applications. Existing empirical defenses, including RLHF, guardrails, and decoding-time interventions, lack formal assurances, limiting their reliability against adversarial exploits. To address this gap, we introduced `C-SafeGen`, a model-agnostic certification framework that provides theoretical risk bounds and enables the enforcement of provable safety constraints. Complemented by our novel CSD decoding algorithm, `C-SafeGen` ensures both safety and fluency in generated text. Empirical evaluations confirm the efficacy of our approach, demonstrating its potential as a robust foundation for the deployment of provably safe LLMs.

## Acknowledgements

This work is partially supported by the National Science Foundation under grant No. 1910100, No. 2046726, NSF AI Institute ACTION No. IIS-2229876, DARPA TIAMAT No. 80321, the National Aeronautics and Space Administration (NASA) under grant No. 80NSSC20M0229, ARL Grant W911NF-23-2-0137, Alfred P. Sloan Fellowship, the research grant from eBay, AI Safety Fund, Virtue AI, and Schmidt Science.

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

# A  Omitted proofs

## A.1  Proof of Theorem 1

*Proof of Theorem 1.* The proof sketch follows [3]. Since the risk function $R(\cdot, \cdot)$ is upper bounded by 1, we can apply a tighter version of Hoeffding's inequality [13] for $\hat{\alpha} > \mathbb{E}[R_G(x, P_\alpha(x))]$:

$$\mathbb{P}\left[R_G(x, P_\alpha(x)) \geq \hat{\alpha}\right] \leq \exp\left\{-N_{\text{cal}}h(\hat{R}(\hat{\mathcal{D}}_{\text{cal}}), \hat{\alpha})\right\} \tag{12}$$

Also, applying Bentkus inequality [7], we have:

$$\mathbb{P}\left[R_G(x, P_\alpha(x)) \geq \hat{\alpha}\right] \leq e\mathbb{P}\left[\text{Bin}(N_{\text{cal}}, \hat{\alpha}) \leq \left\lceil N_{\text{cal}}\hat{R}(\hat{\mathcal{D}}_{\text{cal}})\right\rceil\right] \tag{13}$$

Combining Equations (12) and (13), we have:

$$\mathbb{P}\left[R_G(x, P_\alpha(x)) \geq \hat{\alpha}\right] \leq \min\left(\exp\left\{-N_{\text{cal}}h\left(\hat{R}(\hat{\mathcal{D}}_{\text{cal}}), \hat{\alpha}\right)\right\}, e\mathbb{P}\left[\text{Bin}(N_{\text{cal}}, \hat{\alpha}) \leq \left\lceil N_{\text{cal}}\hat{R}(\hat{\mathcal{D}}_{\text{cal}})\right\rceil\right]\right) \tag{14}$$

Or equivalently, given uncertainty $1 - \delta$, we have:

$$\delta = \min\left(\exp\left\{-N_{\text{cal}}h\left(\hat{R}(\hat{\mathcal{D}}_{\text{cal}}), \hat{\alpha}\right)\right\}, e\mathbb{P}\left[\text{Bin}(N_{\text{cal}}, \hat{\alpha}) \leq \left\lceil N_{\text{cal}}\hat{R}(\hat{\mathcal{D}}_{\text{cal}})\right\rceil\right]\right), \tag{15}$$

which leads to the following by formulating an inverse function:

$$\hat{\alpha} = \min\left\{h^{-1}\left(\frac{\ln(1/\delta)}{N_{\text{cal}}}; \hat{R}(\hat{\mathcal{D}}_{\text{cal}})\right), \Phi_{\text{bin}}^{-1}\left(\frac{\delta}{e}; N_{\text{cal}}, \hat{R}(\hat{\mathcal{D}}_{\text{cal}})\right)\right\} \tag{16}$$

$\square$

## A.2  Proof of Theorem 2

*Proof of Theorem 2.* The proof follows [14]. We consider $|\Lambda|$ independent hypothesis test corresponding to the $|\Lambda|$ Null hypothesis. By the Bonferroni method, each test is performed at a significance level of $\frac{\delta}{|\Lambda|}$. Therefore, The probability of not making a Type I error in a single test is $1 - \frac{\delta}{|\Lambda|}$. The probability of making no Type I error in all $|\Lambda|$ tests is $(1 - \frac{\delta}{|\Lambda|})^{|\Lambda|}$. The probability of making at least one Type I error (i.e., FWER) is the complement of making no Type I errors, which is $1 - (1 - \frac{\delta}{|\Lambda|})^{|\Lambda|} \leq \delta$. Therefore, we prove that the familywise error rate is $\delta$ for Bonferroni correction. Thus, going back to the risk guarantee, we conclude the proof. $\square$

## A.3  Proof of Theorem 3

*Proof of Theorem 3.* Let the output distribution by Algorithm 1 be $p_{\text{ccd}}$. To show the result, we need to show that for any output sequence $y \in \mathcal{Y}$, $p_{\text{ccd}}(y) = p_G(y)$.

We consider the following scenarios respectively: (1) this is an invalid output by the guardrail model: $G(y) = 0$; (2) this a valid output: $G(y) = 1$.

**Scenario (1)**: $G(y) = 0$. According to CCD algorithm, such an undesired claim would always be associated with 1.0 backtrack probability in Line 4 of Algorithm 1, and thus we have $p_{\text{ccd}}(y) = 0$. On the other hand, according to Equation (10), we always have $p_G(y) = 0$ when $G(y) = 0$. Therefore, we have $p_{\text{ccd}}(y) = p_G(y)$ when $G(y) = 0$.

**Scenario (2)**: $G(y) = 1$. In this scenario, we basically would like to validate that the backtrack mechanism in Lines 7-9 of Algorithm 1 does not distort the distribution.

**Part (a)**: To show that, we first validate that the support of output distribution by CCD is identical to that of the restricted output distribution.

For any valid support in the restricted output distribution $y = [\tilde{y}_1, \tilde{y}_2, .., \tilde{y}_N]$, there always exists an unrisky claim decoding path. We consider this simple claim decoding path $\tilde{y}_1, \tilde{y}_2, .., \tilde{y}_N$. Due to

Assumption 4.1, since the complete output sequence $o$ is unrisky, then each claim prefix would also be unrisky (otherwise violating the claim risk cascade assumption). Therefore, this claim decoding path is a valid instantiation of CCD algorithm.

Considering the reverse direction, for any valid support output $y$ by CCD algorithm, it is obvious that $G(y) = 1$ given the acceptance of the last claim, and thus, this output is also a valid support of the restricted output distribution $p_G$.

**Part (b)**: Then, we will prove that for any valid support $y = [\tilde{y}_1, \tilde{y}_2, .., \tilde{y}_N]$ such that $p_G(y) > 0$ and $p_{\mathrm{ccd}}(y) > 0$, we have $p_G(y) = p_{\mathrm{ccd}}(y)$.

For ease of notation, we notate $\alpha = \sum_{y \in \mathcal{Y}} p(y)\mathbb{I}[G(y) = 1]$, which is a normalization factor in Equation (10). Then following the chain rule, we can compute $p_G(y)$ as:

$$p_G(y) = \frac{1}{\alpha} p(\tilde{y}_1) p(\tilde{y}_2|\tilde{y}_1) \cdots p(\tilde{y}_N|\tilde{y}_1, ..., \tilde{y}_{N-1}) \tag{17}$$

Since $y$ is a desired output (i.e., $G(y) = 1$), each claim prefix is also desired according to the risk cascade assumption in Assumption 4.1. Then we have:

$$p_G(y) = \frac{1}{\alpha} p(\tilde{y}_1, G([\tilde{y}_1]) = 1) p(\tilde{y}_2, G([\tilde{y}_1, \tilde{y}_2]) = 1|\tilde{y}_1) \cdots p(\tilde{y}_N, G([\tilde{y}_1, ..., \tilde{y}_N]) = 1|\tilde{y}_1, ..., \tilde{y}_{N-1}) \tag{18}$$

Then we analyze the point mass of output $y$ by the CCD algorithm $p_{\mathrm{ccd}}(y)$. We can formulate the decoding path of CCD algorithm that leads to output $y$ as $\tilde{\boldsymbol{y}}_1, \tilde{y}_1, \tilde{\boldsymbol{y}}_2, \tilde{y}_2, ..., \tilde{\boldsymbol{y}}_N, \tilde{y}_N$, where $\tilde{\boldsymbol{y}}_n$ denotes a sequence of backtracked risky claims at time step $n$.

Then we prove a key invariability property that

$$p(\tilde{y}_n, , G([\tilde{y}_1, ..., \tilde{y}_n]) = 1|\tilde{y}_1, ..., \tilde{y}_{n-1}) = \sum_{\tilde{\boldsymbol{y}}_n} p(\tilde{\boldsymbol{y}}_n, \tilde{y}_n, G([\tilde{y}_1, ..., \tilde{y}_n]) = 1|\tilde{y}_1, ..., \tilde{y}_{n-1}) \tag{19}$$

We can derive as the following:

$$\sum_{\tilde{\boldsymbol{y}}_n} p(\tilde{\boldsymbol{y}}_n, \tilde{y}_n, G([\tilde{y}_1, ..., \tilde{y}_n]) = 1|\tilde{y}_1, ..., \tilde{y}_{n-1}) \tag{20}$$

$$= \sum_{\tilde{\boldsymbol{y}}_n} p(\tilde{\boldsymbol{y}}_n, G([\tilde{y}_1, ..., \tilde{y}_n]) = 1|\tilde{y}_1, ..., \tilde{y}_{n-1}) p(\tilde{y}_n, G([\tilde{y}_1, ..., \tilde{y}_n]) = 1|\tilde{y}_1, ..., \tilde{y}_{n-1}, \tilde{\boldsymbol{y}}_n) \tag{21}$$

$$= \sum_{\tilde{\boldsymbol{y}}_n} p(\tilde{\boldsymbol{y}}_n, G([\tilde{y}_1, ..., \tilde{y}_n]) = 1|\tilde{y}_1, ..., \tilde{y}_{n-1}) p(\tilde{y}_n, G([\tilde{y}_1, ..., \tilde{y}_n]) = 1|\tilde{y}_1, ..., \tilde{y}_{n-1}) \tag{22}$$

For ease of notation, let $u = p(\tilde{y}_n, G([\tilde{y}_1, ..., \tilde{y}_n]) = 0|\tilde{y}_1, ..., \tilde{y}_{n-1})$ and $v = p(\tilde{y}_n, G([\tilde{y}_1, ..., \tilde{y}_n]) = 1|\tilde{y}_1, ..., \tilde{y}_{n-1})$, then we have:

$$\sum_{\tilde{\boldsymbol{y}}_n} p(\tilde{\boldsymbol{y}}_n, G([\tilde{y}_1, ..., \tilde{y}_n]) = 1|\tilde{y}_1, ..., \tilde{y}_{n-1}) p(\tilde{y}_n, G([\tilde{y}_1, ..., \tilde{y}_n]) = 1|\tilde{y}_1, ..., \tilde{y}_{n-1}) \tag{23}$$

$$= (1 - u)v + u(1 - u)v + u^2(1 - u)v + u^3(1 - u)v + \cdots \tag{24}$$

$$= v \tag{25}$$

$$= p(\tilde{y}_n, G([\tilde{y}_1, ..., \tilde{y}_n]) = 1|\tilde{y}_1, ..., \tilde{y}_{n-1}) \tag{26}$$

Therefore, we have:

$$p_{\mathrm{ccd}}(y) = \frac{1}{\alpha} \prod_{n=1}^{N} \sum_{\tilde{\boldsymbol{y}}_n} p(\tilde{\boldsymbol{y}}_n, \tilde{y}_n, G([\tilde{y}_1, ..., \tilde{y}_n]) = 1|\tilde{y}_1, ..., \tilde{y}_{n-1}) \tag{27}$$

$$= \frac{1}{\alpha} \prod_{n=1}^{N} p(\tilde{y}_n, G([\tilde{y}_1, ..., \tilde{y}_n]) = 1|\tilde{y}_1, ..., \tilde{y}_{n-1}) \tag{28}$$

$$= p_G(y) \tag{29}$$

$\square$

# B  Discussion and Limitation

The `C-SafeGen` framework is able to safeguard LLMs' practical deployment and applications against ethical and societal concerns. Existing research shows that the responses of LLMs can be biased towards some demographic groups and not be aligned with human ethics. With `C-SafeGen`, we can define a bias/ethics risk function and control the generation risk below a desired level. The risk guarantee provided by `C-SafeGen` enhances the use of LLMs, addressing societal issues and regulatory infringements. We do not expect any negative societal consequences for our work.

While `C-SafeGen` provides formal guarantees under mild assumptions, it currently relies on the availability of reliable external classifiers or risk estimators to define the safety risk function. The effectiveness of the certification heavily depends on the quality and coverage of these components, which may be incomplete or biased in practice. Additionally, `C-SafeGen` operates in a black-box setting and does not account for internal model behaviors or hidden representations that might influence safety. Finally, although `C-SafeGen` bounds risk on the evaluation distribution, distributional shifts at deployment time (e.g., novel user queries or emerging adversarial strategies) may compromise safety, highlighting the need for future extensions toward adaptive or online certification mechanisms.

