# OpenReview forum: "C-SafeGen: Certified Safe LLM Generation with Claim-Based Streaming Guardrails"
_NeurIPS.cc/2025/Conference — NeurIPS 2025 poster_

### Official Review · Reviewer_F15f · 2025-06-10

**Clarity:** 3
**Significance:** 3
**Originality:** 3
**Rating:** 4
**Confidence:** 4

**Summary:**

Most defenses in the field of LLM adversarial robustness are empirical. The authors address this gap and propose a certified detection and safe decoding defense that is based on conformal analysis. They empirically demonstrate that their derived bounds are tight and that their proposed algorithm improves safety compared to previous methods.

**Questions:**

- What is the proposed threat model? (Evasion attacks?)
- Are there any assumptions made on the distribution of training and test time attacks?
- Are there any assumptions on the expected risk and the function that provides it (e.g., could the classifier be attacked as well and provide wrong estimates?)

**Ethical Concerns:**

["NO or VERY MINOR ethics concerns only"]

**Final Justification:**

As I stated in my initial review, I believe that this is a valuable contribution.
My remaining concerns could be clarified in the rebuttal.

**Limitations:**

yes

**Quality:**

3

**Strengths And Weaknesses:**

Strengths:
* Unification of a detection framework and a safe decoding method
* Derived theoretical bounds appear to be relevant in practice
* Ablation studies demonstrate the importance of individual components
* Methods to improve the runtime are proposed and ablated. The resulting algorithm is reasonably efficient. Specifically, compared to similar approaches

Weaknesses:
* Although I have knowledge about conformal principles, I found the theoretical part hard to follow. I am missing major assumptions that are made to motivate the theory about the expected safety risk etc. The CDF bounds are straightforward, but why are they reasonable in this setting / Why can we assume anything about the expected safety risk (after all, the classifier could be attacked as well and provide a wrong signal)?
* It is not clear to me why we could use the principles of conformal prediction, for example, for evasion attacks. What are the exact threat models this paper is considering?
* I think spending more time to explain the individual components in (3) and (4) and why we can make certain assumptions would strengthen the paper

I generally think this could be a valuable contribution and am very open to improve my score after the rebuttal phase

---

> ### Author Rebuttal · Authors · 2025-07-31
>
> We sincerely thank the reviewer for the thoughtful feedback and constructive suggestions!
>
> > Q1. More clarification on the assumptions and certification in the paper.
>
> Thank you for the valuable feedback! Our paper presents two types of certification. First, we provide **conformal analysis-based certification** for safety risk, captured in Theorem 1 and Theorem 2. Theorem 1 offers a statistical upper bound on the safety risk, while Theorem 2 certifies which configuration meets a target nominal risk level. These results rely on the standard assumption that calibration and test samples are drawn from identical distribution, which also implies that the attack distributions at calibration and inference time should be identical. Notably, we do not make any assumptions about the quality or robustness of the safety risk function itself. If the guardrail model is weak or adversarially attacked, the calibration may yield a looser bound, but the statistical guarantees still hold. Although our work focuses on the i.i.d. setting, the conformal analysis framework can be extended to non-i.i.d. scenarios—such as online or time-series data—as discussed in prior work (e.g., [1]), without changing the core statistical logic.
>
> Second, we provide **algorithmic certification** for our proposed Claim-based Stream Decoding (CSD) algorithm, as formalized in Theorem 3. This result ensures that the CSD algorithm preserves a restricted output distribution that excludes the support of unsafe outputs, as defined in Definition 1. While the determination of safety relies on an external guardrail model, our proof does not assume that this model is accurate or robust. If the guardrail is compromised or poorly calibrated, the resulting restricted distribution may differ from the ideal safe distribution—but the algorithm will still align with the distribution as defined by the given guardrail model. In practice, having a reliable and robust guardrail is crucial for achieving strong empirical performance, but it is not required for the theoretical guarantees of our method.
> [1] Zaffran, Margaux, et al. "Adaptive conformal predictions for time series." International Conference on Machine Learning. PMLR, 2022.
>
> > Q2: Clarification on the threat model.
>
> The primary threat model considered in this paper is the jailbreak setting, where attackers craft adversarial input prompts to elicit unsafe responses from LLMs. Our use of conformal analysis is motivated by the common practice of conducting internal red teaming prior to deployment, which aims to uncover potential jailbreaks. Based on these identified attack distributions, we perform conformal calibration during this pre-deployment stage. This allows us to establish statistical safety guarantees that align with predefined safety policies or regulatory requirements at deployment time.
>
> > Q3: Writing suggestions.
>
> Thank you for the suggestion! We will incorporate these discussions on the threat model and theoretical implications into the paper to enhance clarity and strengthen the overall understanding.

---

> > ### Comment · Reviewer_F15f · 2025-08-01
> > **Thanks for the response and clarifications**
> >
> > I thank the authors for their clarifications. My concerns were sufficiently addressed.
> > I read the concerns raised by the other reviewers and the authors responses.
> > I specifically read the review and Reviewer 1E9p and was satisfied with the author's response.
> > I plan to adjust my score accordingly.

---

### Official Review · Reviewer_1E9p · 2025-06-24

**Clarity:** 1
**Significance:** 1
**Originality:** 2
**Rating:** 4
**Confidence:** 4

**Summary:**

To address the challenge of safe LLM generation with certified guarantees, this paper proposes a Claim-based Stream Decoding (CSD) algorithm coupled with a statistical risk guarantee framework using conformal analysis.
Specifically, CSD utilizes an off-the-shelf Guardrail model to backtrack and resample claim streams if flagged with risks.
This paper also provides theoretical proof of the proposed method.
Experiments against five baselines demonstrate the effectiveness of CSD in safety protection.

**Questions:**

**Questions**:
1. Could you provide a randomized baseline that produces random outputs, in order to estimate a lower-bound threshold for safety?

***
**Typos**:
1. Line 90: Does "SafeGen" refer to "C-SafeGen" or to safe generation in general?
2. Line 108: Should "extitSafeGen" be "SafeGen"?
3. Line 138: Should "below $\alpha$" be corrected to "below $r$"?
4. Line 239: "four baselines" but five are listed.
***

**Ethical Concerns:**

["NO or VERY MINOR ethics concerns only"]

**Final Justification:**

Some of my concerns — specifically weaknesses 3 and 4 — have been addressed.

For weaknesses 1 and 2, I am not fully convinced they have been resolved, but the authors have promised to improve the formulation, enhance the writing, and correct the typos. I trust the authors to deliver on this commitment and have therefore raised my score from 2 to 4.

**Limitations:**

The authors have discussed the limitations in the Appendix.

**Quality:**

2

**Strengths And Weaknesses:**

**Strengths**:
1. This paper addresses the important problem of safety-guaranteed LLM generation.
2. The proposed method CSD, which utilizes an off-the-shelf Guardrail model to backtrack and resample claim streams, is effective at preserving safety according to the experiments.
3. A specific design is introduced to improve the efficiency of CSD by caching previous verified claims.
***
**Weaknesses**:
1. The paper is hard to follow.
    * Some formulations are unnecessary
    * The core idea and design of C-SafeGen are not clearly highlighted. I had to carefully read Algorithm 1 to grasp the full picture.
2. It is unclear how the SafeGen protocol $P_a$  is calculated. Specifically, the meaning of "configuration" in this context is vague. Does it mean the A more detailed explanation of key terms is needed.
3. The experients are not sufficient. For example, there are no experiments to verify potential performance degradation caused by CSD. Without such evaluation, it is unclear whether the safety protection actually works. Additionally, the efficiency comparison lacks baselines such as *Vanilla*.
4. The experimental details are not adequately explained.
    * For evaluation metrics, I can only tell which models are used in Section 5.1.
    * Neither prompts nor open-source code are provided for the function $R_G$.
***

---

> ### Author Rebuttal · Authors · 2025-07-31
>
> We sincerely thank the reviewer for the thoughtful feedback and constructive suggestions!
>
> > Q1: Improvement of writing.
>
> Thank you for the valuable suggestion! We will include an overview figure to illustrate the two types of certification presented in the paper. (1) We provide conformal analysis-based certification for safety risk, as formalized in Theorems 1 and 2. Theorem 1 establishes a statistical upper bound on safety risk, while Theorem 2 certifies configurations that satisfy a target nominal risk level. (2) We also provide algorithmic certification for our proposed Claim-based Stream Decoding (CSD) algorithm, as detailed in Theorem 3. This guarantees that CSD preserves a restricted output distribution that excludes the support of unsafe outputs, as defined in Definition 1. Additionally, we will simplify the notations, provide a notation table, and correct typos to improve clarity and readability.
>
> > Q2: Clarification of the certified configuration.
>
>
> The **configuration** refers to the set of hyperparameters used in our CSD generation process, including the backtrack probability $\alpha_b$, stagnation tolerance $\alpha_g$, and minimum claim length $\alpha_l$. The **certified configuration** $P_\alpha$ is derived according to **Theorem 2**. Specifically, we construct a set of null hypotheses over the safety risks associated with different configurations and retain those with acceptable p-values under **family-wise error rate (FWER) control**, ensuring a bounded overall error rate. We will include additional details in the remark following Theorem 2 for clarity.
>
>
> > Q3: More evaluations on utility and efficiency of the vanilla method.
>
> Thank you for the valuable feedback! In response to your suggestion, we evaluate **Llama-3.1-8B** on **MMLU-Pro** as a benign utility test, shown in Table 1. The results align with our theoretical analysis in **Theorem 3**, which states that CSD generation preserves the restricted output distribution. Since MMLU-Pro contains only benign inputs that do not trigger unsafe responses, CSD does not initiate any backtracking—resulting in **no drop in benign utility**. We would like to emphasize that the CSD algorithm relies on a backtracking mechanism triggered only upon detection of unsafe content. Therefore, when evaluating on fully benign domains, backtracking is rarely invoked—ensuring that utility remains unaffected.
>
>
> In addition, we provide efficiency and safety risk evaluations of the **vanilla method** in **Table 2**, alongside other decoding strategies. While the vanilla method incurs minimal inference cost, it suffers from significantly higher safety risk compared to CSD.
>
> Regarding the evaluation of a randomized baseline as a lower bound on safety, we respectfully disagree that randomized outputs necessarily represent the worst-case (i.e., most unsafe) responses. Random sampling may not consistently produce harmful outputs. We would appreciate further clarification on this suggestion and are open to conducting additional evaluations based on a more detailed formulation.
>
>
>
> **Table 1: MMLU-Pro score of Llama-3.1-8B under vanilla and CSD generation (using LlamaGuard3-8B as the guardrail model)**
>
> | Generation Method | Overall Score |
> |-------------------|---------------|
> | Vanilla           | 0.441         |
> | CSD               | 0.441         |
>
> **Table 2: Evaluation of Decoding Runtime and Mean Safety Risk**
>
> | Method           | Runtime per Instance (s) | Mean Safety Risk |
> |------------------|---------------------------|------------------|
> | Vanilla          | 1.34                      | 0.83             |
> | Best-of-N        | 41.50                     | 0.42             |
> | CSD              | 7.53                      | 0.12             |
> | CSD + KV Cache   | 4.05                      | 0.11             |
>
>
>
>
>
>
>
> > Q4: Clarification of experiment details.
>
> As described in Section 5.1, we use LlamaGuard3-8B and ShieldGemma-9B as the guardrail models for evaluating the safety risk function. We follow the Hugging Face implementations and use the official prompts provided for both models. We will clarify this in Section 5.1 and also explicitly highlight the inference model and the evaluation guardrail model in each table to improve clarity and understanding.

---

> > ### Comment · Reviewer_1E9p · 2025-08-01
> >
> > Thank you for your response. I believe most of my concerns have been addressed, and I will therefore increase my score accordingly.
> >
> > I strongly suggest that the authors carefully revise the paper — particularly to clarify the formulation, improve the writing, and correct any typos — and ensure these revisions are made as promised. Otherwise, the review will not have any real meaning
> >
> > Good luck with your acceptance!

---

### Official Review · Reviewer_qLuo · 2025-07-01

**Clarity:** 3
**Significance:** 2
**Originality:** 2
**Rating:** 5
**Confidence:** 3

**Summary:**

This paper introduces C-SafeGen, a certification framework that provides theoretical safety risk guarantees for large language model (LLM) generations. The framework includes a novel decoding algorithm, Claim-based Stream Decoding (CSD), which uses a guardrail model to monitor claims and claim "segmentation" method, that divides the input prompt into a set of non-overlapping claims. Empirical evaluations show that CSD reduces unsafe generations compared to existing methods.

**Questions:**

1. A clarification question: equation 7 defines a "claim backtrack probability" function, but the defined quantity is an indicator function, not a probability. Can the authors clarify?

2. A question regarding the design choices of the method: Instead of segmenting the output into claims, some alternate approaches might be:
    - cumulative safety checking every "K" generated output tokens by the guardrail model, with backtracking
    - eliminate the claim-based system, and have the model re-generate the output if it is deemed unsafe by the guardrail model

I wonder whether the authors have considered other such simpler design choices for evaluation? Can the authors comment on the trade-offs between different design choices?

**Ethical Concerns:**

["NO or VERY MINOR ethics concerns only"]

**Final Justification:**

The author rebuttals addressed my questions regarding utility of the underlying approach. As a result, I am increasing my score and recommending an accept.

**Limitations:**

Yes

**Quality:**

3

**Strengths And Weaknesses:**

Strengths:
+ The methodology introduced by this paper is simple, effective and theoretically sound. It involves using guardrail models that already exist, but instead of using these to filter unsafe outputs; this paper proposes to use these to create an alternate decoding mechanism to enable models to create safe outputs instead, instead of simply blocking the generation of an unsafe one.
+ The experimental results are sound. The proposed approach lowers the safety risk of outputs while being more computationally efficient compared to approaches such as best-of-N.
+ The writing is clear, concise, and to the point.

Weaknesses:
- The main weakness in this paper is a lack of evaluations of utility of the resulting safe outputs. In principle, it is trivial to certify safety: simply refuse to produce an output, or produce the same benign output (e.g.: "I cannot answer this question") for every prompt. The main question is whether we can certify safety while ensuring that the utility of the LLM does not deteriorate significantly. While Theorem 3 in the paper argues for this theoretically, empirical evaluations on this end would help better motivate this approach.

- Another weakness of this paper is the usage of a rather simple claim-point detection method. For simplicity, the paper chooses to detect claims by identifying newlines, periods, etc. This has several obvious failure cases, for example, when a single long sentence may have multiple claims, or a claim being spread across sentences. Thus this choice appears to be ad-hoc, and the lack of discussion around this topic is a weakness.

---

> ### Author Rebuttal · Authors · 2025-07-31
>
> We sincerely thank the reviewer for the thoughtful feedback and constructive suggestions!
>
> > Q1: Additional utility evaluation of CSD algorithm.
>
> Thank you for the valuable feedback! In response to your suggestion, we evaluate **Llama-3.1-8B** on **MMLU-Pro** as a benign utility test, shown in Table 1. The results align with our theoretical analysis in **Theorem 3**, which states that CSD generation preserves the restricted output distribution. Since MMLU-Pro contains only benign inputs that do not trigger unsafe responses, CSD does not initiate any backtracking—resulting in **no drop in benign utility**. We would like to emphasize that the CSD algorithm relies on a backtracking mechanism triggered only upon detection of unsafe content. Therefore, when evaluating on fully benign domains, backtracking is rarely invoked—ensuring that utility remains unaffected.
>
> **Table 1: MMLU-Pro score of Llama-3.1-8B under vanilla and CSD generation (using LlamaGuard3-8B as the guardrail model)**
>
> | Generation Method | Overall Score |
> |-------------------|---------------|
> | Vanilla           | 0.441         |
> | CSD               | 0.441         |
>
> > Q2: Discussion on claim-point detection methods.
>
> Thank you for raising this insightful point! We agree that if a language model uses a different set of claim termination tokens than the ones we define, the effectiveness of claim-based generation could be compromised. In such cases, it may be necessary to tune the termination token set accordingly. We will add this clarification in our revision.
>
> We also appreciate your suggestions on alternative ways to define claims and incorporate safety checks. Specifically:
>
> 1. **Fixed-length claims**: Treat every *k* tokens as a claim and perform safety checks and potential backtracking after each segment.
> 2. **Full-response claims**: Treat the entire response as a single claim and perform a post hoc safety check with potential resampling.
>
> Option 2 corresponds to **Best-of-N sampling**, which we have already evaluated. As shown in our results, CSD significantly outperforms Best-of-N under the same computational budget. For Option 1, we believe it may lead to inefficiencies and suboptimal performance due to local semantic dependencies. For example, if *k = 5* and the model generates a safe first segment like “Here is how to make,” the next segment is likely to complete the phrase in an unsafe way (e.g., “bombs: ...”) due to the semantic trajectory of the generation. This illustrates why **semantically meaningful termination points** are important for effective and efficient safety checks.
>
> Motivated by this, we adopt a **claim-based decoding** strategy, where claims are semantically grounded units that serve as natural checkpoints for safety validation and potential backtracking. We will clarify this design motivation in our updated version.
>
>
>
> > Q3: Confusion on the backtrack probability function.
>
> Apologies for the confusion. We will revise the name to **claim backtrack decision function** and update **Algorithm 1** accordingly to ensure consistency and clarity.

---

> > ### Comment · Reviewer_qLuo · 2025-08-06
> >
> > Thank you for the rebuttal!
> >
> > Re: the utility evaluation of the CSD algorithm, I expect there might be a lot of corner cases (e.g.: "write a linux command to kill a process") where evaluation is a nuanced question, but I understand that text in standard benchmarks is often safe. Still, it is good to know that the current algorithm does not have obvious failure cases.
> >
> > Thanks also for the discussion around claim-point detection. I believe discussing these aspects in the main paper will help potential readers.
> >
> > As my questions have been addressed, I shall raise my score accordingly.

---

### Official Review · Reviewer_NPt4 · 2025-07-02

**Clarity:** 3
**Significance:** 2
**Originality:** 3
**Rating:** 4
**Confidence:** 2

**Summary:**

This paper proposes a Claim-based Stream Decoding (CSD) algorithm coupled with a statistical risk guarantee framework using conformal analysis. CSD algorithm integrates a stream guardrail model to safeguard sequential claims generated by LLMs and incorporates a backtracking mechanism to revise claims flagged with high safety risks.  Empirical evaluations demonstrate the effectiveness and efficiency of the CSD algorithm compared to baselines.

**Questions:**

Could the authors show on how sensitive results are to the choice of the threshold $\alpha_{b}$ and $\alpha_{g}$?

**Ethical Concerns:**

["NO or VERY MINOR ethics concerns only"]

**Final Justification:**

After reading other reviews, there are still a few concerns unresolved. It is uncertain how the final version looks like. I therefore maintain my score.

**Limitations:**

yes

**Quality:**

2

**Strengths And Weaknesses:**

Strengths:

1. The paper is clearly written and well-structured.

2. The methodology is well-developed and theoretically grounded.


Weaknesses:
1.  Could the authors elaborate more about when Assumption 4.1 hold and when it fails in the real world?

2. The paper focuses on safety rates but does not quantitatively evaluate the effect of CSD on output utility. While it is stated that CSD “ensures fluency and coherence” and preserves “high-quality outputs”, no evaluations are provided. It would be important to measure generation utility after enforcing safety.

3. The experiments focus on two “jailbreak” style benchmarks and adversarial suffix attacks. It is unclear how C-SafeGen would perform on other safety domains. The generalizability beyond the tested scenarios is somewhat uncertain. I suggest the authors involve a discussion or providing examples on other potential safety domains.

---

> ### Author Rebuttal · Authors · 2025-07-31
>
> We sincerely thank the reviewer for the thoughtful feedback and constructive suggestions!
>
> > Q1: When does Assumption 4.1 hold or fail in real-world settings?
>
> Assumption 4.1 states that if a generation contains an unsafe statement, then any continuation of that statement remains unsafe. This assumption generally holds in many real-world cases involving safety-sensitive topics. For example, if an LLM generates bomb-making instructions, the entire response is considered unsafe regardless of what follows—even if the model later includes an apology—because the harmful content remains accessible.
>
> However, this assumption may not always hold, particularly in cases involving nuanced or context-dependent risks such as hate speech. For instance, if a model initially generates a harmful statement but then immediately retracts it with something like, “I said something wrong, and my actual point is...,” the overall response might be deemed safe under certain criteria.
>
> As discussed in the paper, Assumption 4.1 represents a **conservative stance** on safety. It errs on the side of caution by treating any continuation of an unsafe segment as unsafe, which aligns with our goal of enforcing strict safety guarantees in high-risk domains.
>
>
>
> > Q2: Additional utility evaluation of CSD algorithm.
>
> Thank you for the valuable feedback! In response to your suggestion, we evaluate **Llama-3.1-8B** on **MMLU-Pro** as a benign utility test, shown in Table 1. The results align with our theoretical analysis in **Theorem 3**, which states that CSD generation preserves the restricted output distribution. Since MMLU-Pro contains only benign inputs that do not trigger unsafe responses, CSD does not initiate any backtracking—resulting in **no drop in benign utility**. We would like to emphasize that the CSD algorithm relies on a backtracking mechanism triggered only upon detection of unsafe content. Therefore, when evaluating on fully benign domains, backtracking is rarely invoked—ensuring that utility remains unaffected.
>
> **Table 1: MMLU-Pro score of Llama-3.1-8B under vanilla and CSD generation (using LlamaGuard3-8B as the guardrail model)**
>
> | Generation Method | Overall Score |
> |-------------------|---------------|
> | Vanilla           | 0.441         |
>
> > Q3: Generalization to other safety domains.
>
> C-SafeGen is **guardrail-agnostic** and can be readily extended to domains such as misinformation and toxicity, provided that an appropriate guardrail evaluator is available. In fact, the AdvBench and JailbreakBench datasets used in our evaluation already encompass a diverse range of safety risk categories. We will include more category-specific results in the final version to better illustrate C-SafeGen's effectiveness across different risk types.
>
> > Q4: Ablation studies of $\alpha_b$ and $\alpha_g$.
>
> Thank you for the suggestion! We already include a sensitivity analysis of the backtrack threshold $\alpha_b$in **Figure 3**, which demonstrates that increasing the ease of backtracking effectively reduces safety risk.
>
> In addition, we present an ablation study of the **stagnation tolerance** parameter $\alpha_g$ in **Table 2**. The results show that stricter stagnation tolerance (i.e., smaller $\alpha_g$) leads to improved safety performance under the same sampling budget.
>
> **Table 2: Ablation of \(\alpha_g\) on AdvBench with Llama-3.1-8B (guardrail: LlamaGuard3-8B)**
> | $\\alpha_g$                  | 64    | 128   | 256   | $\infty$ (no safe sampling) |
> |------------------------------|-------|--------|--------|-------------------------------|
> | Conformal Safety Risk        | 0.200 | 0.210  | 0.215  | 0.293                         |

---

> > ### Comment · Reviewer_NPt4 · 2025-08-05
> >
> > Thanks for answering my questions.

---

### Official Review · Reviewer_Mjhm · 2025-07-04

**Clarity:** 3
**Significance:** 3
**Originality:** 2
**Rating:** 4
**Confidence:** 4

**Summary:**

This paper deals with the safety guardrail of LLM during inference time. To fill the gap of lacking rigorous safety guarantees in previous approaches, the authors propose a claim-based streaming decoding algorithm with risk control by conformal analysis. This involves a stream guardrail model and a claim partition model with backtracking for improved efficiency. The authors also provide theoretical guarantees by deriving the upper bound of safety risk. Empirical experimental results showcase the efficiency and efficacy of the proposed approach.

**Questions:**

1.	Do you have any ablation for the safe resampling techniques? And which one of the introduced techniques (random masking/permutation/noise injection/structured permutation) is the most effective and efficient?
2.	How is the ShieldGemma-9B implemented and used in the evaluation? ShieldGemma can only make a binary decision for one given class in one inference.

**Ethical Concerns:**

["NO or VERY MINOR ethics concerns only"]

**Final Justification:**

My only unsolved concern is from the evaluation of the limited attacks. Even though the authors claim they evaluated the strongest known white-box jailbreak technique and would incorporate results from black-box jailbreak methods in the revised version, I do not see any additional evaluation results in the current rebuttal. Regardless, I am fine if the paper gets accepted

**Limitations:**

yes

**Quality:**

2

**Strengths And Weaknesses:**

## Strengths
1.	The risk control and safety guarantees of LLM generation during inference time are significant.
2.	The theoretical guarantee is solid with a derived upper bound that allows better control.
3.	The introduced claim separation and KV cache for guardrail models improve the efficiency of inference.
4.	The paper is clearly written and easy to follow.

## Weaknesses:
1.	Reliance on the output probability of guardrail models may not be reliable enough. Existing work [1] found that guard models like Llama-Guard3 tend to be overconfident and miscalibrated under jailbreak attacks, limiting the safety estimation in the framework.
2.	Limited attacks are investigated except for GCG. HarmBench [2] includes multiple types of attacks that may help investigate the effectiveness of the proposed method under different attacks.

## References
[1] Liu, H., Huang, H., Gu, X., Wang, H., & Wang, Y. (2025). On calibration of LLM-based guard models for reliable content moderation. ICLR. \
[2] Mazeika, M., Phan, L., Yin, X., Zou, A., Wang, Z., Mu, N., ... & Hendrycks, D. (2024). Harmbench: A standardized evaluation framework for automated red teaming and robust refusal. ICML.

---

> ### Author Rebuttal · Authors · 2025-07-31
>
> We sincerely thank the reviewer for the thoughtful feedback and constructive suggestions!
>
> > Q1: Discussion on using the output probability of guardrail models as safety risk.
>
> Thank you for highlighting this interesting work! We agree that improved calibration of output probabilities can lead to a more accurate safety risk function and further enhance the effectiveness of our SafeGen framework. We will include a discussion of this direction in our revision.
>
> [1] Liu, H., Huang, H., Gu, X., Wang, H., & Wang, Y. (2025). *On calibration of LLM-based guard models for reliable content moderation*. ICLR.
>
>
>
> > Q2: Evaluations for different safe resampling techniques and additional attack types.
>
> Thank you for the suggestion! We conduct an ablation study to evaluate the impact of each individual safe resampling technique used in safe sampling. We examine the effect of removing each of the following components in Table 1. We observe that excluding any of the individual techniques leads to an increase in conformal safety risk, indicating that each component contributes to overall safety. The full combination of all techniques achieves the lowest risk.
> For attacks, we use GCG, the strongest known white-box jailbreak technique. In our revised version, we will also incorporate results from black-box jailbreak methods to broaden the evaluation scope.
>
> Table 1: Ablation Study on Safe Resampling Techniques on AdvBench with Llama3-8B and LlamaGuard3-8B as guardrail evaluator.
>
> | Safe Resampling Configuration     | Conformal Safety Risk |
> |----------------------------------|------------------------|
> | Excluding Random Masking         | 0.232                  |
> | Excluding Random Permutation     | 0.235                  |
> | Excluding Gaussian Noise Injection | 0.213                |
> | All Techniques Combined (Full CSD)| **0.210**             |
>
>
> > Q3: Implementation details of ShieldGemma-9B.
>
> We follow the official Hugging Face implementation of ShieldGemma-9B, incorporating its four risk categories into the prompt template as the `safety_policy` input. This enables the model to perform safe/unsafe judgments based on those predefined categories with one inference. We will clarify this setup in the evaluation section of the revised version.

---

> > ### Comment · Reviewer_Mjhm · 2025-08-06
> >
> > Thank the authors for the response. After reading other reviews, I would like to keep the original score.

---

### Decision · Program_Chairs · 2025-09-17

**Decision:**

Accept (poster)

**Comment:**

This paper develops a guardrail for LLM safety with a novel Claim-based Stream Decoding (CSD) algorithm. The proposed method enables the model to create safe outputs instead of simply blocking unsafe outputs, with statistical risk guarantees. It also comes with a specific design for improving the efficiency of inference. The contributions of this paper are significant, and both theoretical and empirical results are sound. All the reviewers gave positive ratings after the rebuttal. Therefore, the AC is recommending to accept this paper. The authors are still suggested to improve the clarity of their writing in the revision.